# Clinical impact of broad- versus narrow-spectrum empiric therapy in acute cholangitis: A Japanese claims database study

Kazuhiro Aoto[1,2], Ryo Inose[1], Yuichi Muraki[1]*

1 Laboratory of Clinical Pharmacoepidemiology, Kyoto Pharmaceutical University, 5 Misasaginakauchicho, Kyoto Yamashina-ku, Kyoto, Japan, 2 Department of Pharmacy, University Hospital, Kyoto Prefectural University of Medicine, Kajii-Cho, Kawaramachi-Hirokoji, Kamigyo-Ku, Kyoto, Japan

* y-muraki@mb.kyoto-phu.ac.jp

## Abstract

The clinical benefit of broad-spectrum empiric therapy in patients with acute cholangitis is unclear. We aimed to evaluate the impact of broad-spectrum and narrow-spectrum empiric therapies on patient outcomes using a Japanese claims database. The study included patients who were diagnosed with acute cholangitis between April 2014 and August 2022, aged 18–99 years, received antibiotics, had blood cultures collected, and underwent biliary drainage. Patients who received empiric therapy with carbapenems, piperacillin/tazobactam, or fourth-generation cephalosporins were included in the broad-spectrum group, whereas others were included in the narrow-spectrum group. Of the 4,755 eligible patients, 3,377 were categorized into the narrow-spectrum group and 1,378 into the broad-spectrum group. In the multivariate logistic regression analysis, older age, higher Charlson Comorbidity Index, the presence of sepsis, and intensive care unit admission were associated with increased 30-day in-hospital mortality, whereas the receipt of broad-spectrum empiric therapy was not (adjusted odds ratio, 1.37 [95% confidence interval {CI}, 0.84–2.23]). In the propensity score matching analysis, there was also no association between the receipt of broad-spectrum empiric therapy and 30-day in-hospital mortality (odds ratio, 1.43 [95% CI, 0.82–2.50]). Furthermore, in the propensity score-matched cohort, the broad-spectrum group had longer duration of intravenous antibiotic therapy (median interquartile range [IQR]: 8 [6 –11] day vs. 9 [7 –13] day; difference 1 day [95% CI, 0.31–1.69 day]) and length of hospital stay (median [IQR]: 13 [9 –20] day vs. 16 [11 –25] day; difference 3 day [95% CI, 1.87–4.13 day]), compared with the narrow-spectrum group. In this large-scale study using a Japanese claims database, broad-spectrum empiric therapy was not associated with improved clinical outcomes, compared with narrow-spectrum empiric therapy. Therefore, the necessity of broad-spectrum empiric therapy may be limited in clinical practice, and narrow-spectrum empiric therapy may represent an effective treatment strategy for acute cholangitis.

**Data availability statement:** Data availability: Data used in this study are owned by JMDC Inc. This data cannot be made publicly available due to the terms of license agreement between JMDC Inc. and Kyoto Pharmaceutical University. The authors have no special access privileges that others would not have. If other researchers wish to use the data, they need to purchase it directly from JMDC Inc. For access requests, please contact JMDC Inc. via its website (https://www.jmdc.co.jp/en/).

**Funding:** The author(s) received no specific funding for this work.

**Competing interests:** RI received funding for the commissioned research from Kowa Company, Ltd. However, this study was not directly funded. YM received grants from Pfizer Japan Inc., Kowa Company Ltd., the Japan Pharmaceutical Association, the Japan Society for the Promotion of Science, and the Ministry of Health, Labour and Welfare. YM is also a board member of the Japanese Society of Pharmaceutical Health Care and Sciences and the Japanese Society for Infection Prevention and Control, and a committee member of the AMR Clinical Reference Center. However, this study was not directly funded. KA declares no conflict of interest. This does not alter our adherence to PLOS ONE policies on sharing data and materials.

## Introduction

Acute cholangitis is frequently complicated by bacteremia, and some patients develop septic shock or organ dysfunction [1–4]. Antimicrobial therapy and biliary drainage are the key treatments for acute cholangitis that improve patient outcomes [5].

The major pathogens that cause acute cholangitis are *Escherichia coli* and *Klebsiella pneumoniae* [1]. Over the recent decades, antimicrobial resistance among these *Enterobacteriales* has become a global public health concern [6–8]. To address the potential presence of these antimicrobial resistance, clinical guidelines for acute cholangitis recommend empiric therapy with carbapenems, piperacillin/tazobactam, or fourth-generation cephalosporins [9,10]. These broad-spectrum antibiotics are also positioned as empiric therapy targeting *Pseudomonas aeruginosa* [9,10], which is rarely isolated as a causative pathogen of acute cholangitis [1].

The use of broad-spectrum antibiotics in empiric therapy is a common treatment strategy for infections; however, recent clinical reports have highlighted their overuse [11–13]. Such overuse not only contributes to the spread of antimicrobial resistance but may also be associated with higher mortality, an increased risk of *Clostridioides difficile* infection (CDI), and prolonged length of hospital stay (LOS) [11,14–17]. Consequently, the consistent benefits of broad-spectrum empiric therapy on patient outcomes have been questioned, and the overuse of broad-spectrum antibiotics in empiric therapy has been recognized as an issue that needs to be addressed [11,16–18].

To date, it remains unclear whether broad-spectrum empiric therapy is associated with better outcomes than narrow-spectrum empiric therapy in acute cholangitis. A retrospective cohort study showed that patient outcomes did not improve with broad-spectrum empiric therapy compared to narrow-spectrum empiric therapy, suggesting that the benefit of broad-spectrum empiric therapy is limited [19]. However, this was a single-center study with a small sample size. To our knowledge, no large-scale cohort study has evaluated the benefits of broad-spectrum versus narrow-spectrum empiric therapy on clinical outcomes in patients with acute cholangitis.

Healthcare databases enable comprehensive data collection from large patient populations across multiple facilities and have therefore been widely used to evaluate medication efficacy [20]. Particularly, claims databases have recently been used for studies on antimicrobial therapy in large populations [17,21].

This study aimed to examine whether broad-spectrum empiric therapy is associated with improved 30-day in-hospital mortality compared with narrow-spectrum empiric therapy using a Japanese claims database. Furthermore, as a secondary objective, the impact of broad-spectrum empiric therapy on the incidence of CDI, duration of intravenous (IV) antibiotic therapy, and LOS was evaluated.

## Materials and methods

### Study design and data source

This retrospective cohort study was conducted using a hospital database provided by JMDC, Inc. (Tokyo, Japan) [22]. The database contains claims data for all patients

who visited approximately 500 hospitals. This represents approximately 7% of all hospitals in Japan and contains data on approximately 17 million patients. The database includes information on patient demographics, diagnoses, prescriptions, and procedures.

Data extraction from the database began on January 23, 2025. In this study, we used only anonymized administrative claims database and did not have access to information that could identify individual patients. This study was confirmed by the ethics committee of Kyoto Pharmaceutical University as it did not require an ethical review, on January 22, 2025 (NR-00009). This study followed the Reporting of Studies Conducted using Observational Routinely-Collected Data for Pharmacoepidemiology reporting guidelines [23] (S1 Checklist).

## Population

We enrolled patients diagnosed with acute cholangitis between April 2014 and August 2022. The database used in this study contained only monthly diagnostic data and did not include detailed daily diagnostic information. Therefore, to enhance the reliability of the acute cholangitis diagnosis, the following inclusion criteria were adopted: age 18–99 years, first episode of acute cholangitis, receiving antibiotics listed in the Tokyo Guidelines 2018 for more than 2 days [10], blood culture obtained on the day of antibiotic initiation, and biliary drainage performed within 2 days of antibiotic initiation. Patients with unknown discharge dates were excluded as their outcomes could not be assessed.

## Exposures and outcomes

The eligible patients were categorized into broad-spectrum and narrow-spectrum groups. The broad-spectrum group included patients who received carbapenems, piperacillin/tazobactam, or fourth-generation cephalosporins, whereas all others were included in the narrow-spectrum group (S1 Table). The primary outcome was the 30-day in-hospital mortality. The secondary outcomes included the incidence of CDI during hospitalization, duration of IV antibiotic therapy, and LOS. CDI was evaluated as a representative adverse event associated with antimicrobial therapy [24]. The proportion of patients in each group who received broad-spectrum antibiotics as definitive therapy was also investigated.

## Collected data and definitions

The database used in this study did not contain information on culture results. Therefore, antibiotic therapy initiated on the day of blood culture collection was considered empiric therapy, and the date was defined as the index date. The following data were collected based on the index date: age; sex; acquisition type (community-acquired or healthcare-associated); Charlson Comorbidity Index (CCI) [25]; presence of sepsis; vasopressor prescription; intensive care unit (ICU) admission; concomitant prescription of vancomycin, history of immunosuppressant or antibiotic prescription within 90 days; hospital bed count; treatment year. Community-acquired cholangitis was defined as cases in which empiric therapy was initiated within 2 days of admission, whereas healthcare-associated cholangitis was defined as cases in which empiric therapy was initiated more than 2 days after admission [26]. The CCI was collected as a measure of comorbidity burden to assess the impact of comorbidities on prognosis [27].

Diagnoses were identified using the International Classification of Diseases, 10th edition codes. Antibiotics were identified using anatomical therapeutic chemical codes, and blood culture collection was identified using the Japanese procedure code. Definitive therapy was defined as the antibiotic administered on the day of completion of IV antibiotic therapy. All codes used for collection in this study are summarized in S1 and S2 Tables.

## Statistical analysis

Univariate and multivariate logistic regression analyses were performed to evaluate the association between receipt of broad-spectrum empiric therapy and 30-day in-hospital mortality. The exposure variable was the receipt of broad-spectrum

 

empiric therapy (yes/no), and the outcome was 30-day in-hospital mortality (yes/no). Similar to a previous study [28], the multivariable model was adjusted for all variables with a p value ≤ 0.1 in the univariable analysis. Multicollinearity among covariates was assessed by calculating the variance inflation factor for each variable, with values less than 10 considered to indicate no concerning multicollinearity [29]. The association between receipt of broad-spectrum empiric therapy and 30-day in-hospital mortality was also evaluated using a conditional logistic regression model after propensity score (PS) matching. PS matching analysis was performed to adjust for the differences in baseline characteristics between the groups [30]. A nearest neighbor 1:1 matching with a maximum caliper of 0.2 times the standard deviation was applied in the PS matching. The PS was estimated by logistic regression using the following variables: age, sex, acquisition type, CCI, presence of sepsis, vasopressor prescription, ICU admission, history of immunosuppressant or antibiotic prescription within 90 days, hospital bed count, and treatment year. After PS matching, the balance of variables between the groups was evaluated using absolute standardized differences, with differences < 0.1 interpreted as well balanced.

The incidence of CDI, duration of IV antibiotic therapy, and LOS were reported as between-group differences after PS matching; bootstrapping with 1,000 samples was used for 95% confidence intervals (CIs). Categorical variables were expressed as absolute numbers, and continuous variables were expressed as medians and interquartile ranges (IQRs). Significance in all statistical analyses was defined as a two-sided p value < 0.05. All statistical analyses were performed using the Stata software (version 18.0; StataCorp LLC., College Station, TX, USA).

### Sensitivity analysis

Sensitivity analyses were performed to evaluate the robustness of the primary results. First, to rule out the possibility that blood cultures were not collected on the day of antimicrobial therapy initiation in severely ill patients, we examined the baseline characteristics and 30-day in-hospital mortality rates in patients without blood culture collection. Second, a sensitivity analysis restricted to patients with sepsis in the PS-matched cohort was conducted to assess the robustness of the association between broad-spectrum empiric therapy and 30-day in-hospital mortality.

## Results

### Patient characteristics

The study cohort included 182,407 patients diagnosed with acute cholangitis (Fig 1). Of these, 4,755 patients were eligible, with 3,377 categorized into the narrow-spectrum group and 1,378 into the broad-spectrum group. Table 1 shows the baseline characteristics of the patients in each group, and Table 2 lists the antibiotics included in each group. A larger proportion of patients with sepsis was found in the broad-spectrum group (n = 489 [35.5%]) than in the narrow-spectrum group (n = 743 [22.0%]) (Table 1).

Cefoperazone/sulbactam (n = 2,011 [59.6%]) was the most commonly administered antibiotic in the narrow-spectrum group, followed by cefmetazole (n = 663 [19.6%]) (Table 2). In contrast, piperacillin/tazobactam (n = 813 [59.0%]) was the most common antibiotic in the broad-spectrum group, followed by meropenem (n = 465 [33.7%]). The 30-day in-hospital mortality rates in the narrow- and broad-spectrum groups were 1.3% (n = 43) and 2.2% (n = 31), respectively. Supporting Information S1 File shows the results of a sensitivity analysis conducted in patients without blood culture collection.

### Logistic regression analysis of variables associated with 30-day in-hospital mortality

Table 3 shows the variables associated with 30-day in-hospital mortality in the logistic regression analysis. In multivariate logistic regression analysis, older age (adjusted odds ratio [aOR], 1.06 [95% CI, 1.03–1.09], p < 0.001), higher CCI (aOR, 1.32 [95% CI, 1.19–1.47], p < 0.001), the presence of sepsis (aOR, 1.89 [95% CI, 1.17–3.06], p = 0.010), and ICU admission (aOR, 2.65 [95% CI, 1.26–5.58], p = 0.010) were significantly associated with increased 30-day in-hospital mortality, whereas receipt of broad-spectrum empiric therapy was not significantly (aOR, 1.37 [95% CI, 0.84–2.23], p = 0.20). Variance inflation factor values ranged from 1.01 to 1.17, indicating no concerning collinearity of the variables.

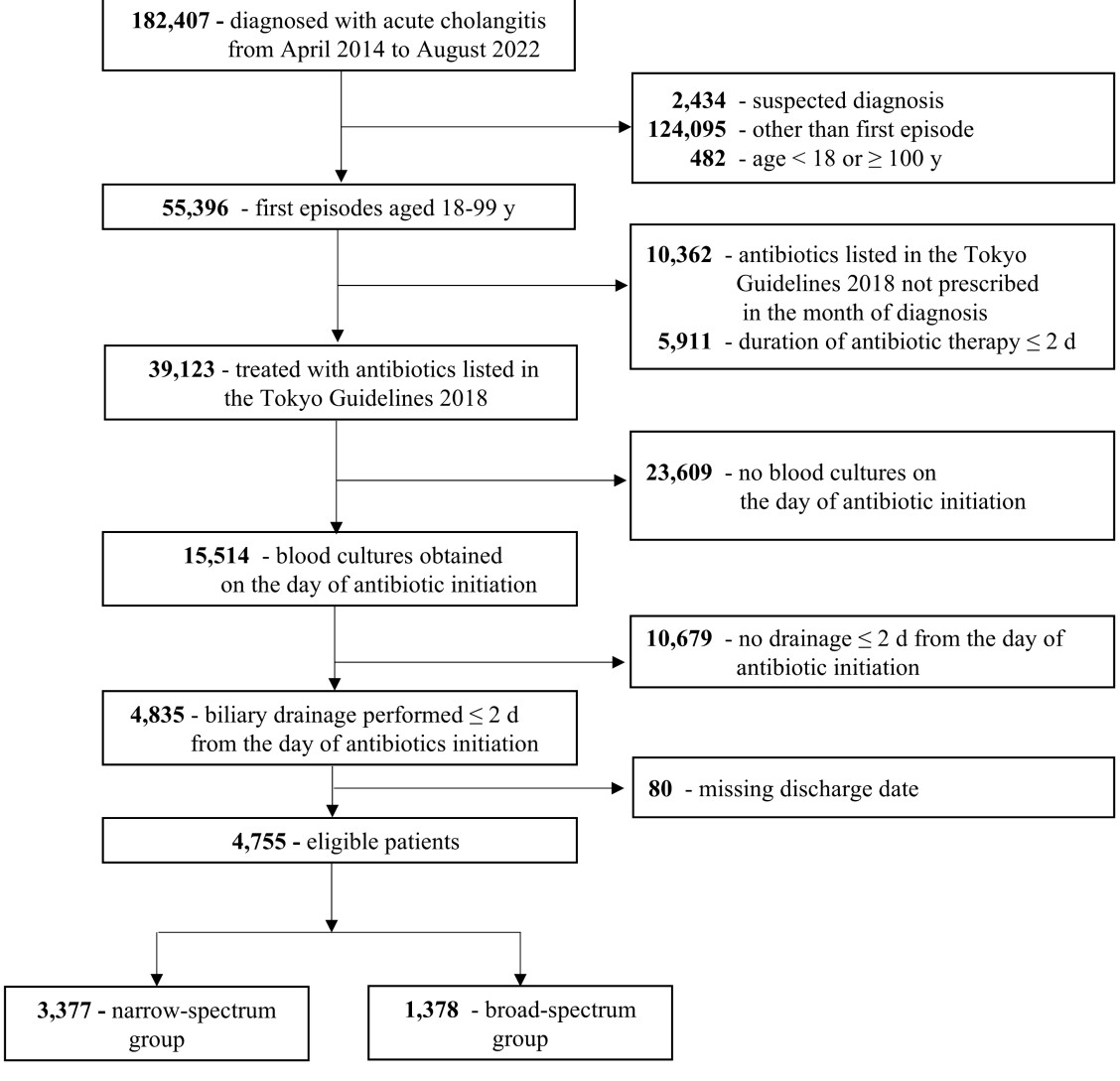

**Fig 1. Flow diagram of the patient selection process.**

## Patient outcomes in the PS-matched cohort

After PS matching, 1,378 patients in each group were included, and all baseline variables were well-balanced (S3 Table). The antibiotics included were similar to those before the PS matching analysis (S4 Table). A conditional logistic regression model showed that receipt of broad-spectrum empiric therapy was not significantly associated with 30-day in-hospital mortality (OR, 1.43 [95% CI, 0.82–2.50], p = 0.21). Furthermore, no significant association was observed even in a sensitivity analysis restricted to patients with sepsis (OR, 1.48 [95% CI, 0.71–3.12], p = 0.30).

Table 4 presents the results regarding the incidence of CDI during hospitalization, duration of IV antibiotic therapy, and LOS. The incidence of CDI did not significantly differ between the narrow-and broad-spectrum groups (5 [0.36%] vs. 8 [0.58%], difference [95% CI]: 0.22 [−0.32–0.75]%, p = 0.43). In contrast, compared with the narrow-spectrum group, the broad-spectrum group had longer duration of IV antibiotic therapy (median [IQR]: 8 [6 −11] day vs. 9 [7 −13] day;

**Table 1. Baseline characteristics of the patients.**

| | Narrow-spectrum group (n = 3377) | Broad-spectrum group (n = 1378) |
|---|---|---|
| **Demographics** | | |
| **Male sex** | 1864 (55.2) | 753 (54.6) |
| **Age, y, median (IQR)** | 81 (73–87) | 82 (75–88) |
| **Age ≥ 75 y** | 2418 (71.6) | 1058 (76.8) |
| **Community-acquired cholangitis** | 3187 (94.4) | 1254 (91.0) |
| **CCI, median (IQR)** | 1 (0–2) | 1 (0–2) |
| **Sepsis** | 743 (22.0) | 489 (35.5) |
| **Vasopressor prescription** | 162 (4.8) | 87 (6.3) |
| **ICU admission** | 106 (3.1) | 106 (7.7) |
| **History of prescriptions** | | |
| Immunosuppressant(s) | 259 (7.7) | 161 (11.7) |
| Antibiotic(s) | 401 (11.9) | 201 (14.6) |
| **Combination antibiotics** | | |
| Vancomycin | 6 (0.2) | 6 (0.4) |
| **Hospital bed count** | | |
| ≤ 199 | 221 (6.5) | 109 (7.9) |
| 200–499 | 1957 (58.0) | 668 (48.5) |
| ≥ 500 | 1199 (35.5) | 601 (43.6) |
| **Treatment year** | | |
| 2014 | 80 (2.4) | 23 (1.7) |
| 2015 | 138 (4.1) | 34 (2.5) |
| 2016 | 167 (4.9) | 57 (4.1) |
| 2017 | 220 (6.5) | 70 (5.1) |
| 2018 | 499 (14.8) | 218 (15.8) |
| 2019 | 590 (17.5) | 211 (15.3) |
| 2020 | 656 (19.4) | 267 (19.4) |
| 2021 | 699 (20.7) | 317 (23.0) |
| 2022 | 328 (9.7) | 181 (13.1) |

Data are presented as numbers (%) unless otherwise indicated.

Abbreviations: CCI, Charlson Comorbidity Index; ICU, intensive care unit; IQR, interquartile range.

difference [95% CI]: 1 [0.31–1.69] day, p = 0.004) and LOS (median [IQR]: 13 [9 –20] day vs. 16 [11 –25] day; difference [95% CI]: 3 [1.87–4.13], p < 0.001).

As definitive therapy, 87.3% (n = 1,203) of the patients in the narrow-spectrum group continued receiving narrow-spectrum antibiotics, and only 10.7% (n = 147) required escalation to broad-spectrum antibiotics (S1 Fig). In contrast, in the broad-spectrum group, although 34.8% (n = 480) of the patients were de-escalated to narrow-spectrum antibiotics, 63.2% (n = 871) continued to receive broad-spectrum antibiotics.

## Discussion

This study showed that administration of broad-spectrum empiric therapy was not associated with improved 30-day in-hospital mortality. This finding was consistent across both the logistic regression and PS matching analyses.

**Table 2. Antibiotics included in the narrow-spectrum and broad-spectrum groups.**

| Group | Classification | Antibiotics | n (%) |
|---|---|---|---|
| **Narrow-spectrum group** | | | **3377 (100)** |
| | Combinations of penicillins and beta-lactamase inhibitors | Ampicillin/sulbactam | 463 (13.71) |
| | First-generation cephalosporins | Cefazolin | 12 (0.36) |
| | Second-generation cephalosporins | Cefmetazole | 663 (19.63) |
| | | Cefotiam | 37 (1.10) |
| | | Flomoxef | 7 (0.21) |
| | Third-generation cephalosporins | Cefoperazone/sulbactam | 2011 (59.55) |
| | | Ceftriaxone | 147 (4.35) |
| | | Cefotaxime | 9 (0.27) |
| | | Ceftazidime | 4 (0.12) |
| | Fluoroquinolones | Ciprofloxacin | 2 (0.06) |
| | | Levofloxacin | 21 (0.62) |
| | Monobactams | Aztreonam | 1 (0.03) |
| **Broad-spectrum group** | | | **1378 (100)** |
| | Fourth-generation cephalosporins | Cefepime | 14 (1.02) |
| | | Cefozopran | 13 (0.94) |
| | Combinations of penicillins and beta-lactamase inhibitors | Piperacillin/tazobactam | 813 (59.00) |
| | Carbapenems | Meropenem | 465 (33.74) |
| | | Doripenem | 62 (4.50) |
| | | Imipenem/cilastatin | 11 (0.80) |

**Table 3. Univariate and multivariate logistic analyses of variables associated with 30-day in-hospital mortality.**

| Variables | Univariate analysis | | Multivariate analysis | |
|---|---|---|---|---|
| | OR (95% CI) | p value[a] | aOR (95% CI) | p value[a] |
| **Broad-spectrum empiric therapy** | 1.78 (1.12–2.84) | 0.015 | 1.37 (0.84–2.23) | 0.20 |
| **Male sex** | 0.81 (0.51–1.29) | 0.38 | | |
| **Age, y** | 1.04 (1.01–1.07) | 0.002 | 1.06 (1.03–1.09) | < 0.001 |
| **Community-acquired cholangitis** | 0.50 (0.25–1.02) | 0.057 | 0.59 (0.29–1.22) | 0.16 |
| **CCI** | 1.33 (1.22–1.46) | < 0.001 | 1.32 (1.19–1.47) | < 0.001 |
| **Sepsis** | 2.21 (1.39–3.52) | 0.001 | 1.89 (1.17–3.06) | 0.010 |
| **Vasopressor prescription** | 0.50 (0.12–2.04) | 0.33 | | |
| **ICU admission** | 3.05 (1.50–6.22) | 0.002 | 2.65 (1.26–5.58) | 0.010 |
| **History of immunosuppressant prescription** | 2.68 (1.51–4.77) | 0.001 | 1.64 (0.84–3.21) | 0.15 |
| **History of antibiotic prescription** | 2.25 (1.32–3.86) | 0.003 | 1.59 (0.87–2.92) | 0.13 |

Abbreviations: aOR, adjusted odds ratio; CCI, Charlson Comorbidity Index; CI, confidence interval; ICU, intensive care unit; OR, odds ratio.

[a]Significance level: p = 0.05.

Furthermore, sensitivity analysis restricted to patients with sepsis showed similar results, supporting the robustness of the results. In addition, in the PS-matched cohort, patients in the broad-spectrum group had longer duration of IV antibiotic therapy and LOS (Table 4). Collectively, the present study did not identify any clear clinical benefits of broad-spectrum empiric therapy. To the best of our knowledge, this is the largest study to evaluate the impact of broad-spectrum empiric therapy for acute cholangitis.

**Table 4. Incidence of CDI, duration of IV antibiotic therapy, and LOS in the PS-matched cohort.**

| Outcomes | No. of events (%) or median (IQR) | | Difference, % or days (95% CI) | p value[e] |
|---|---|---|---|---|
| | Narrow-spectrum group (n = 1378) | Broad-spectrum group (n = 1378) | | |
| CDI | 5 (0.36)[a] | 8 (0.58)[a] | 0.22 (−0.32–0.75)[b] | 0.43 |
| Duration of IV antibiotic therapy | 8 (6–11)[c] | 9 (7–13)[c] | 1 (0.31–1.69)[d] | 0.004 |
| LOS | 13 (9–20)[c] | 16 (11–25)[c] | 3 (1.87–4.13)[d] | < 0.001 |

Abbreviations: CDI, Clostridioides difficile infection; CI, confidence interval; IQR, interquartile range; IV, intravenous; LOS, length of hospital stay.

[a]Data are presented as numbers (%).

[b]Data are presented as % (95% CI).

[c]Data are presented as median (IQR).

[d]Data are presented as days (95% CI).

[e]Significance level: p = 0.05.

Piperacillin/tazobactam and meropenem, which are generally active against third-generation cephalosporin-resistant *Enterobacterales* and *P. aeruginosa* [31–33], were commonly used in broad-spectrum empiric therapy (Table 2). Although blood culture results were unavailable in this study, the prevalence of third-generation cephalosporin resistance among *E. coli* and *K. pneumoniae*, the main pathogens of acute cholangitis [1], is approximately 10–20% in Japan [32]. In addition, the isolation rate of *P. aeruginosa* in acute cholangitis is low, at approximately 1–2% [1]. These epidemiological data suggest that a large proportion of patients with acute cholangitis in Japan can be treated without broad-spectrum antibiotics. Indeed, this study showed that escalation to broad-spectrum antibiotics as definitive therapy was uncommon in the narrow-spectrum group, suggesting that most patients were adequately treated with narrow-spectrum antibiotics (S1 Fig).

In contrast, despite the lower likelihood of resistant pathogens in Japan, 63.2% of the patients in the broad-spectrum group continued to receive broad-spectrum antibiotics without de-escalation. This finding implies that de-escalation may not be sufficiently implemented in patients with acute cholangitis who received broad-spectrum empiric therapy. A previous study indicated that de-escalation was less likely when culture results were negative [34]. Considering that the positive blood culture rate for acute cholangitis is approximately 40% [1], some patients in the broad-spectrum group may not have undergone de-escalation because their blood cultures were negative. Furthermore, the continuation of broad-spectrum antibiotics may be influenced by physicians' strong sense of reassurance regarding the efficacy of these agents, as has been reported in studies on antibiotic prescription behavior [35,36].

This study also showed that the duration of IV antibiotic therapy and LOS were longer in the broad-spectrum group versus the narrow-spectrum group (Table 4). Our previous single-center study showed that narrow-spectrum empiric therapy is associated with shorter IV antibiotic therapy and LOS, through active switching to oral antibiotics [19]. These findings suggest that measures are needed to avoid unnecessary prolonged use of antibiotics, particularly in patients who have received broad-spectrum empiric therapy. Although broad-spectrum empiric therapy has been reported to be associated with an increased incidence of CDI [11,16], no significant difference was observed between the groups in this study (Table 4). This result was probably because the incidence was lower than that reported in previous studies [11,16].

Nevertheless, de-escalation and the duration of IV antibiotic therapy may be influenced not only by the physician's psychology but also by patient-related factors, including individual clinical background [37,38]. Although this study adjusted for patient background using PS matching analysis, it could not fully account for these factors. Therefore, our findings do not imply that broad-spectrum empiric therapy is always unnecessary for the treatment of acute cholangitis. Rather, these results suggest that broad-spectrum empiric therapy for acute cholangitis could be an important focus of antimicrobial stewardship, highlighting the need for more judicious indications.

This study has some limitations. First, to address the confounders, we performed multivariate logistic regression and PS matching analyses. However, disease severity may not have been adequately adjusted for because the database did not contain patients' vital parameters, laboratory results, and detailed organ dysfunction measures. In addition, information regarding the diagnosis and treatment at institutions outside the database could not be collected and was unknown. Therefore, residual confounding may exist based on the indications and severity. Second, only patients who had undergone biliary drainage were included in this study to enhance the reliability of the acute cholangitis diagnosis. Consequently, our cohort may have included a higher proportion of patients with favorable prognoses. Early and successful source control via drainage may offset the mortality-reducing effects of broad-spectrum empiric therapy, limiting its generalizability to situations in which drainage is delayed or unavailable. Additionally, it was unclear whether source control was successfully achieved after drainage. Differences in the achievement of source control may have influenced patient outcomes in both groups. Third, our findings may have been influenced by the low prevalence of antimicrobial resistance in Japan. Broad-spectrum empiric therapy may be beneficial for patient outcomes in regions with a high prevalence of antimicrobial resistance. Further research is warranted to evaluate the impact of broad-spectrum empiric therapy in such settings. Despite these limitations, this study provides new insights into the role of broad-spectrum antibiotics in empiric therapy for acute cholangitis.

## Conclusion

In this study, broad-spectrum empiric therapy was not associated with improved clinical outcomes compared with narrow-spectrum empiric therapy. This finding, based on large-scale Japanese claims data, suggests that the necessity of broad-spectrum empiric therapy is limited and that narrow-spectrum empiric therapy may represent an effective treatment strategy for acute cholangitis. The active use of narrow-spectrum antibiotics as empiric therapy can help reduce the use of broad-spectrum antibiotics and may contribute to the prevention of antimicrobial resistance.

## Supporting information

**S1 Checklist. The reporting of studies conducted using observational routinely-collected data for pharmacoepidemiology checklist.**
(PDF)

**S1 Table. Antibiotics included in the analysis.**
(DOCX)

**S2 Table. Codes used for variable definitions.**
(DOCX)

**S1 File. Supporting information. A sensitivity analysis in the patient selection process.**
(DOCX)

**S3 Table. Baseline characteristics of the patients before and after PS matching.**
(DOCX)

**S4 Table. Antibiotics included in the narrow-spectrum and broad-spectrum groups after PS matching.**
(DOCX)

**S1 Fig. Proportions of antibiotics in the categories used as definitive therapy stratified by empiric therapy.**
(PDF)

## Acknowledgments

We would like to thank Editage (www.editage.jp) for the English language editing.

## Author contributions

**Conceptualization:** Kazuhiro Aoto.

**Data curation:** Kazuhiro Aoto.

**Formal analysis:** Kazuhiro Aoto.

**Investigation:** Kazuhiro Aoto.

**Methodology:** Kazuhiro Aoto.

**Project administration:** Ryo Inose, Yuichi Muraki.

**Resources:** Yuichi Muraki.

**Software:** Kazuhiro Aoto.

**Supervision:** Yuichi Muraki.

**Validation:** Kazuhiro Aoto.

**Visualization:** Kazuhiro Aoto.

**Writing – original draft:** Kazuhiro Aoto.

**Writing – review & editing:** Ryo Inose, Yuichi Muraki.

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
