## [Decision Letter · Decision Letter 0]

11 Jan 2026

PONE-D-25-61122Clinical impact of broad- versus narrow-spectrum empiric therapy in acute cholangitis: A Japanese claims database studyPLOS One

Dear Dr. Muraki,

Thank you for submitting your manuscript to PLOS ONE. After careful consideration, we feel that it has merit but does not fully meet PLOS ONE’s publication criteria as it currently stands. Therefore, we invite you to submit a revised version of the manuscript that addresses the points raised during the review process. 

We look forward to receiving your revised manuscript.

Kind regards,

Csaba Varga, DVM MSc PhD

Academic Editor

PLOS One

Journal Requirements:

RI received funding for commissioned research from the Kowa Company, Ltd. However, this study was not directly funded. YM received grants from Pfizer Japan Inc.; Kowa Company Ltd.; the Japan Pharmaceutical Association; the Japan Society for the Promotion of Science; and the Ministry of Health, Labour and Welfare. YM is also a board member of the Japanese Society of Pharmaceutical Health Care and Sciences and the Japanese Society for Infection Prevention and Control, and a committee member of the AMR Clinical Reference Center. However, this study was not directly funded. KA has no conflicts of interest to declare.

4. Please remove all personal information, ensure that the data shared are in accordance with participant consent, and re-upload a fully anonymized data set.

Reviewers' comments:

Reviewer's Responses to Questions

**Comments to the Author**

1. Is the manuscript technically sound, and do the data support the conclusions?

Reviewer #1: Yes

Reviewer #2: Yes

2. Has the statistical analysis been performed appropriately and rigorously? 

Reviewer #1: Yes

Reviewer #2: Yes

3. Have the authors made all data underlying the findings in their manuscript fully available?

The PLOS Data policy requires authors to make all data underlying the findings described in their manuscript fully available without restriction, with rare exception (please refer to the Data Availability Statement in the manuscript PDF file). The data should be provided as part of the manuscript or its supporting information, or deposited to a public repository. For example, in addition to summary statistics, the data points behind means, medians and variance measures should be available. If there are restrictions on publicly sharing data—e.g. participant privacy or use of data from a third party—those must be specified.requires authors to make all data underlying the findings described in their manuscript fully available without restriction, with rare exception (please refer to the Data Availability Statement in the manuscript PDF file). The data should be provided as part of the manuscript or its supporting information, or deposited to a public repository. For example, in addition to summary statistics, the data points behind means, medians and variance measures should be available. If there are restrictions on publicly sharing data—e.g. participant privacy or use of data from a third party—those must be specified.requires authors to make all data underlying the findings described in their manuscript fully available without restriction, with rare exception (please refer to the Data Availability Statement in the manuscript PDF file). The data should be provided as part of the manuscript or its supporting information, or deposited to a public repository. For example, in addition to summary statistics, the data points behind means, medians and variance measures should be available. If there are restrictions on publicly sharing data—e.g. participant privacy or use of data from a third party—those must be specified.requires authors to make all data underlying the findings described in their manuscript fully available without restriction, with rare exception (please refer to the Data Availability Statement in the manuscript PDF file). The data should be provided as part of the manuscript or its supporting information, or deposited to a public repository. For example, in addition to summary statistics, the data points behind means, medians and variance measures should be available. If there are restrictions on publicly sharing data—e.g. participant privacy or use of data from a third party—those must be specified.

Reviewer #1: Yes

Reviewer #2: Yes

4. Is the manuscript presented in an intelligible fashion and written in standard English?

Reviewer #1: Yes

Reviewer #2: Yes

5. Review Comments to the Author

Reviewer #1: This study, Clinical impact of broad- versus narrow-spectrum empiric therapy in acute cholangitis: A Japanese claims database study, is a retrospective study utilizing a significant Japanese claims database to address a critical question in antimicrobial stewardship. The authors demonstrate that, within their matched cohort, broad-spectrum empiric therapy was not associated with improved 30-day in-hospital mortality. However, the study’s reliance on administrative data creates several structural limitations regarding clinical intent and disease severity that require more rigorous exploration through sensitivity analyses to confirm the robustness of the findings.

1. Justification of exclusions (selection bias)

The patient selection process excluded 23,609 patients because they did not have blood cultures collected on the day of antibiotic initiation. This represents a significant portion of the original population.

From the reviewer’s knowledge, it is crucial to determine if clinicians systematically omit blood cultures in less severe cases or, conversely, in cases so critical that immediate intervention precludes culture collection.

- Please provide a sensitivity analysis comparing the baseline characteristics and 30-day mortality of the 23,609 excluded patients with the 4,755 included patients.

2. Sensitivity analysis of the "sepsis" subgroup

The study correctly identifies that sepsis was significantly more prevalent in the broad-spectrum group (35.5% vs. 22.0%).

While Propensity Score (PS) matching was used, the ‘benefit’ of broad-spectrum agents is often most pronounced in the highest-acuity patients. A subgroup-specific analysis would confirm if the "no benefit" finding holds true for the most vulnerable patients.

- Perform a sensitivity analysis restricted only to patients diagnosed with sepsis at the index date.

3. The length of hospital stay (LOS) and intravenous (IV) duration

The broad-spectrum group had significantly longer hospital stays (16 days vs. 13 days) and IV therapy duration (9 days vs. 8 days).

Since 63.2% of the broad-spectrum group failed to de-escalate, the prolonged LOS may be tied more to stewardship failures than drug performance.

- Discuss whether this is a biological consequence of antibiotic efficacy or a behavioral consequence of clinician conservative bias when treating patients they perceive as high-risk.

4. Causal interpretation, confounding, and generalizability

Given the observational design and limitations of claims data, readers could over‑interpret the findings as definitive evidence that broad‑spectrum empiric therapy is unnecessary in all cases of acute cholangitis. The manuscript would benefit from more precise framing of the target population and more explicit discussion of residual confounding and regional microbiology.

Please explicitly acknowledge:

- The likelihood of residual confounding by indication and severity, despite multivariable adjustment and PS matching, due to the absence of physiologic data, laboratory tests, and detailed organ dysfunction measures.

- That early and successful source control via drainage may diminish any incremental mortality benefit of broader empiric coverage, which limits generalizability to settings where drainage is delayed or less accessible.

- That the observed low prevalence of third‑generation cephalosporin resistance in E. coli and Klebsiella spp. and the low isolation rate of Pseudomonas in cholangitis in Japan may not hold in regions with higher resistance burdens, where empiric decisions may appropriately differ.

Lastly, please consider softening the definitive statements in the Conclusion of ‘broad-spectrum empiric therapy showed no clinical benefit’.

Reviewer #2: This study addresses an important and clinically relevant question regarding empiric antibiotic selection in patients with acute cholangitis. The use of a large, nationwide Japanese claims database provides a robust sample size and enhances the generalizability of the findings within the studied healthcare setting. The authors employ appropriate and well-established statistical methods, including multivariable logistic regression and propensity score matching, to account for confounding in this observational design. The inclusion of clinically meaningful outcomes, such as 30-day in-hospital mortality and healthcare utilization measures, further strengthens the study. However, the manuscript would benefit from a more comprehensive review of the existing literature and greater clarity in the definition and explanation of key exposure and outcome measures, which would improve contextualization of the findings and overall interpretability. Overall, the study contributes valuable evidence to ongoing discussions around antimicrobial stewardship in the management of acute cholangitis.

6. PLOS authors have the option to publish the peer review history of their article (what does this mean?). If published, this will include your full peer review and any attached files.). If published, this will include your full peer review and any attached files.). If published, this will include your full peer review and any attached files.). If published, this will include your full peer review and any attached files.

...

Reviewer #1: No

Reviewer #2: **Yes:** Isha AgrawalIsha AgrawalIsha AgrawalIsha Agrawal

---

## [Author Response · Author response to Decision Letter 1]

16 Feb 2026

The reviewers' responses are included in the manuscript as a separate file. Please refer to that file.

---

## [Decision Letter · Decision Letter 1]

19 Mar 2026

Clinical impact of broad- versus narrow-spectrum empiric therapy in acute cholangitis: A Japanese claims database study

PONE-D-25-61122R1

Dear Dr. Yuichi Muraki,

We’re pleased to inform you that your manuscript has been judged scientifically suitable for publication and will be formally accepted for publication once it meets all outstanding technical requirements.

Kind regards,

Csaba Varga, DVM MSc PhD

Academic Editor

PLOS One

Additional Editor Comments (optional):

Reviewers' comments:

Reviewer's Responses to Questions

**Comments to the Author**

1. If the authors have adequately addressed your comments raised in a previous round of review and you feel that this manuscript is now acceptable for publication, you may indicate that here to bypass the “Comments to the Author” section, enter your conflict of interest statement in the “Confidential to Editor” section, and submit your "Accept" recommendation.

Reviewer #1: All comments have been addressed

Reviewer #2: All comments have been addressed

2. Is the manuscript technically sound, and do the data support the conclusions?

Reviewer #1: Yes

Reviewer #2: Yes

3. Has the statistical analysis been performed appropriately and rigorously? 

Reviewer #1: Yes

Reviewer #2: Yes

4. Have the authors made all data underlying the findings in their manuscript fully available?

The PLOS Data policy requires authors to make all data underlying the findings described in their manuscript fully available without restriction, with rare exception (please refer to the Data Availability Statement in the manuscript PDF file). The data should be provided as part of the manuscript or its supporting information, or deposited to a public repository. For example, in addition to summary statistics, the data points behind means, medians and variance measures should be available. If there are restrictions on publicly sharing data—e.g. participant privacy or use of data from a third party—those must be specified.requires authors to make all data underlying the findings described in their manuscript fully available without restriction, with rare exception (please refer to the Data Availability Statement in the manuscript PDF file). The data should be provided as part of the manuscript or its supporting information, or deposited to a public repository. For example, in addition to summary statistics, the data points behind means, medians and variance measures should be available. If there are restrictions on publicly sharing data—e.g. participant privacy or use of data from a third party—those must be specified.requires authors to make all data underlying the findings described in their manuscript fully available without restriction, with rare exception (please refer to the Data Availability Statement in the manuscript PDF file). The data should be provided as part of the manuscript or its supporting information, or deposited to a public repository. For example, in addition to summary statistics, the data points behind means, medians and variance measures should be available. If there are restrictions on publicly sharing data—e.g. participant privacy or use of data from a third party—those must be specified.requires authors to make all data underlying the findings described in their manuscript fully available without restriction, with rare exception (please refer to the Data Availability Statement in the manuscript PDF file). The data should be provided as part of the manuscript or its supporting information, or deposited to a public repository. For example, in addition to summary statistics, the data points behind means, medians and variance measures should be available. If there are restrictions on publicly sharing data—e.g. participant privacy or use of data from a third party—those must be specified.

Reviewer #1: Yes

Reviewer #2: Yes

5. Is the manuscript presented in an intelligible fashion and written in standard English?

Reviewer #1: Yes

Reviewer #2: Yes

6. Review Comments to the Author

Reviewer #1: The revised manuscript substantially improves the clarity and contextualization of the study by sharpening the research question, integrating current evidence on antimicrobial resistance and acute cholangitis, and clearly delineating primary and secondary objectives.

Reviewer #2: Thank you for carefully addressing the reviewer comments in the revised manuscript. The authors have made substantial improvements, and the manuscript is now much clearer and stronger. The previous concerns have been adequately addressed, and I appreciate the revisions made to the text. In its current form, the manuscript is in good shape and suitable for publication.

One minor suggestion:

Line 229: The incidence of CDI did not "significantly" differ: was missing significantly as the values do differ.

Line 231-234: Its not clear which numbers are for broad-spectrum and which ones for narrow. To me the 1st ones are broad and 2nd ones are narrow, however, based on your explanation, it's the opposite. Please rephrase the lines to clarify it.

7. PLOS authors have the option to publish the peer review history of their article (what does this mean?). If published, this will include your full peer review and any attached files.). If published, this will include your full peer review and any attached files.). If published, this will include your full peer review and any attached files.). If published, this will include your full peer review and any attached files.

...

Reviewer #1: No

Reviewer #2: **Yes:** Isha AgrawalIsha AgrawalIsha AgrawalIsha Agrawal

---

## [Editor Report · Acceptance letter]

PONE-D-25-61122R1

PLOS One

Dear Dr. Muraki,

I'm pleased to inform you that your manuscript has been deemed suitable for publication in PLOS One. Congratulations! Your manuscript is now being handed over to our production team.

Kind regards,

on behalf of

Dr. Csaba Varga

Academic Editor

PLOS One